# Optimizing Antibiotic Prescribing for Acute Respiratory Tract Infections in German Primary Care: Results of the Regional Intervention Study CHANGE-3 and the Nested cRCT

**DOI:** 10.3390/antibiotics12050850

**Published:** 2023-05-04

**Authors:** Gregor Feldmeier, Christin Löffler, Attila Altiner, Anja Wollny, Katharina Garbe, Dorothea Kronsteiner, Martina Köppen, Joachim Szecsenyi, Mirko Leyh, Arwed Voss, Martina Kamradt, Regina Poß-Doering, Michel Wensing, Petra Kaufmann-Kolle

**Affiliations:** 1Institute of General Practice, Rostock University Medical Center, 18055 Rostock, Germany; gregor.feldmeier@med.uni-rostock.de (G.F.);; 2Department of General Practice and Health Services Research, University Hospital Heidelberg, 69120 Heidelberg, Germany; 3Institute of Medical Biometry, University Hospital Heidelberg, 69117 Heidelberg, Germany; 4aQua-Institute for Applied Quality Improvement and Research in Health Care GmbH, 37073 Göttingen, Germany; 5Department of Communication Design and Media, University of Applied Sciences, Technology, Business and Design, 23966 Wismar, Germany

**Keywords:** antibacterial agents, respiratory tract infection, drug prescriptions, primary care, physician-patient relation, health communication, antibiotic resistance

## Abstract

Within primary care, acute respiratory tract infections (ARTIs) are the most common reason for prescribing antibiotics. The aim of the CHANGE-3 study was to investigate how antibiotic prescribing for non-complicated ARTIs can be reduced to a reasonable level. The trial was conducted as a prospective study consisting of a regional public awareness intervention in two regions of Germany and a nested cluster randomised controlled trial (cRCT) of a complex implementation strategy. The study involved 114 primary care practices and comprised an intervention period of six winter months for the nested cRCT and two times six winter months for the regional intervention. The primary outcome was the percentage of antibiotic prescribing for ARTIs between baseline and the two following winter seasons. The regression analysis confirmed a general trend toward the restrained use of antibiotics in German primary care. This trend was found in both groups of the cRCT without significant differences between groups. At the same time, antibiotic prescribing was higher in routine care (with the public campaign only) than in both groups of the cRCT. With regard to secondary outcomes, in the nested cRCT, the prescribing of quinolones was reduced, and the proportion of guideline-recommended antibiotics increased.

## 1. Introduction

The increase in antibiotic resistance is one of the fundamental challenges of our time. Despite reductions in recent decades, antibiotics are still used too often and too carelessly. First and foremost, prescribing antibiotics in primary care, which accounts for 80–90% of all prescriptions in human medicine [1,2], has a great savings potential. Acute respiratory tract infections (ARTIs) are by far the most common reason for prescribing antibiotics in primary care [3,4,5]. This is surprising because ARTIs are often caused by viruses, and therefore, treatment with antibiotics is usually not indicated. Nevertheless, in Germany, about 40% of all ARTI cases seen in primary care are still treated with antibiotics [6,7]. It is noteworthy that Southern European countries such as Greece or Italy have higher total consumption rates, while rates in the Netherlands and Austria, for example, are lower than in Germany [6,7]. 

One explanation for General Practitioners (GPs) prescribing “against better knowledge” is the presumed patient expectation regarding antibiotics, which is regularly overestimated by physicians [8]. In addition, broad-spectrum antibiotics (e.g., quinolones and cephalosporins) are prescribed too often due to the (irrational) perception of greater safety [2]. 

Numerous initiatives and studies, especially in the past 15 years, have investigated how prescribing rates for ARTIs could be further reduced [9,10,11,12,13,14]. Today, we have enough evidence for the effectiveness of educational interventions, especially when they are conveyed through reflection on one’s own actions as well as collegial exchange, e.g., in quality circles [15]. Furthermore, there is a considerably good evidence base for two other interventional approaches: firstly, changing doctor–patient communication towards shared decision-making [16] and secondly, influencing indication and drug choice through data-based prescription feedback [17,18]. Previous studies on shared decision-making have mostly focused on the prescribing physician as the initiator of a shared decision-making process [16]. Regarding the importance of practice team involvement, there is conflicting evidence on reducing inappropriate antibiotic prescribing. Some studies have found that by improving workflow, non-physician staff can reduce the burden on the physician and thus improve medical decision-making [19,20]. Moreover, regional and national campaigns have led to a reduction in ambulatory antibiotic use in several European countries [7]. Which effects of a combination of regional and practice-specific intervention might have, has not yet been the subject of research.

The aim of the CHANGE-3 study was to investigate how antibiotic prescribing for non-complicated ARTIs in primary care can be reduced to a reasonable level in two regional areas of Germany. The regional intervention made use of a public awareness campaign, including web-based activities (https://www.weniger-antibiotika.de/, accessed on 3 May 2023), written information for GPs, posters for waiting rooms, patient magazines, a plush toy, colouring booklets, and comics for children. Within the regional intervention, a cluster randomised controlled trial (cRCT) was nested to investigate whether additional interventions generated further effects. The cRCT included practice-specific feedback on antibiotic prescribing, educational outreach visits, recommendations for and e-learning on physician–patient communication, and advanced patient information (flyers, tablet PCs). 

## 2. Results

### 2.1. Baseline Data cRCT

In the nested cRCT, data from 61,390 cases from 114 practices were analysed from T0 to T3, with 27,408 cases from 57 practices in the intervention group and 33,982 cases from 57 practices in the control group. Thus, the number of cases in the intervention group was lower than the number of cases in the control group. There were fluctuations between the individual points of measurement, whereby participating patients were newly included at each point of measurement. Even though patients could be included at multiple time points, there was no follow-up at the individual patient level. In both groups and at all points of measurement, the proportion of cases from Baden-Württemberg was higher (68.8–72.8%) than the proportion of cases from Mecklenburg-Western Pomerania. Most of the patients were between 18 and 65 years old, and the proportion of female patients slightly dominated in both groups. Just over half of the patients had no increased disease burden in terms of comorbidity (hierarchised morbidity groups, HMG = 0), around 30% of the patients suffered from one or two chronic diseases, and—depending on the group and point of measurement—approximately 15–20% of the patients suffered from more than three chronic diseases and participated in a Disease Management Program. In both groups, most patients considered were diagnosed with an ARTI (74.6–82.7%); fewer patients suffered from sinusitis (16.7–22.5%), tonsilitis (3.3–5.7%), or otitis media (3.2–5.2%). The percentage of antibiotic prescribing decreased in the patients considered in both groups (from 23.7% (T0) to 15.1% (T3) in the intervention group and from 17.8% (T0) to 12.4% (T3) in the control group), but the baseline value was significantly higher in the intervention group than in the control group. The participating practices in both groups were individual rather than group practices. See Table 1. 

### 2.2. Antibiotic Prescribing for ARTIs in the cRCT

The primary logistic mixed regression model showed that antibiotic prescriptions for ARTIs declined in both study groups over the study period (T0–T3) (OR = 0.836; *p* < 0.001). However, neither a significant effect between study groups (OR = 1.285; *p* = 0.202) nor a significant interaction effect between time and study group was observed (OR 0.958; *p* = 0.063). See Table 2. The logistic mixed regression model adjusting for additional covariates confirmed the results of the previous model. Again, there was an effect of time (OR = 0.842; *p* < 0.001), but neither a significant effect of the study group (OR = 1.253; *p* = 0.242) nor of an interaction effect between time and study group (OR = 0.956; *p* = 0.053). The model showed that the odds of being prescribed an antibiotic for ARTIs were higher in women than in men (OR = 1.26; *p* < 0.001). Moreover, patients aged 65+ years were more likely to be prescribed an antibiotic for ARTIs than patients aged 18–65 years (OR = 1.297; *p* < 0.001). The odds also increased with the number of hierarchised morbidity groups, i.e., for patients with five or more chronic diseases, the odds were twice as high as for patients without chronic diseases (OR = 2.046; *p* < 0.001). Compared to patients treated in group practices, patients in single practices were less likely to be prescribed an antibiotic (OR = 0.817; *p* = 0.002). See Table 3. 

### 2.3. Baseline Data of the Regional Intervention Group

Table 4 summarises the baseline data of patients who received care as usual and belonged to the regions that were provided with information on optimised antibiotic prescribing as part of the public campaign. In total, data from 4,070,998 cases were considered. At all time points, most of them came from Baden-Württemberg (93.3% at T0). More than 80% of the patients belonged to the age group 18–65 years, while about 10% were younger or older (at T0 82.1% vs. 9.4% and 8.5%, respectively). The study group comprised slightly more women than men (52.4% women at T0). Furthermore, just over half of all patients included in this study group were cared for in practice with only one active GP (54.2% at T0). Most patients were treated for ARTIs, while about a quarter was treated for bronchitis (73.5% and 25.7% at T0). Moreover, there were fewer treatments due to tonsilitis (6.4%), sinusitis (5.2%), and otitis (3.3%). At T0, 25.7% of the patients in this study group were prescribed an antibiotic. The proportion decreased over time to 20.1% at T3. See Table 4. 

### 2.4. Antibiotic Prescribing for ARTIs in Public Campaign Regions

The mixed effects logistic regression model for the primary endpoint is presented in Table 5. First, it shows that in routine care, the percentage of antibiotic prescribing for the index diagnosis decreased over time (OR = 0.903; *p* < 0.001). The comparison of the percentage of antibiotic prescribing in routine care with both groups of the nested cRCT shows that the prescriptions in the control group vs. care as a usual group were significantly lower (OR = 0.634; *p* = 0.001). At the same time, the comparison with the intervention group was not significant (OR = 0.768; *p* = 0.052). The interaction between group and time showed significantly lower antibiotic prescribing in the control and intervention groups compared to care as usual (OR = 0.913; *p* < 0.001, and OR = 0.945, *p* < 0.001, respectively). The effects of gender, age, HMGs, and practice type on antibiotic prescribing are comparable to those of the mixed logistic regression model for the cRCT.

### 2.5. Secondary Outcomes

Quinolone prescribing in the cRCT was already at a low level at T0 (6.8% in the intervention group, 5.3% in the control group) and further decreased to 3.0% and 2.9% at T3. The same was observed for the regional intervention, although more quinolones were prescribed at baseline (9.4% at T0 to 4.2% at T3). In the cRCT, the percentage of prescribing of recommended antibiotics increased, while the percentages remained at the same level in the regional intervention.

## 3. Materials and Methods

### 3.1. Study Design

CHANGE-3 was conducted as a prospective (non-blinded) study consisting of a regional public awareness campaign in two different regions of Germany (Mecklenburg-Western Pomerania, Baden-Württemberg) and a nested cRCT in the named regions. The study comprised an intervention period of six winter months for the nested cRCT (T2) and two times six winter months for the regional intervention (T2 and T3). The outcome evaluation for all study groups (including public awareness campaign) was based on quarterly claims data (§§ 295, 300, Social Code V) provided by two statutory health insurance providers (AOK Nordost, AOK Baden-Württemberg) of the included regions. Primary outcome was the percentage of antibiotic prescribing for ARTIs between baseline (both T0 and T1) and T3.

### 3.2. Recruitment

Regarding the regional intervention, AOK-insured patients with ARTIs from all practicing physicians in both regions were included. The following procedure was chosen for the cRCT: In the fourth quarter of 2017 and the first quarter of 2018, all GPs residents within a 100 km radius of Heidelberg (Baden-Württemberg) and Rostock (Mecklenburg-Western Pomerania) were identified from the complete register of all GPs in each of the two federal states. Random samples were drawn from this population, and the selected GPs were contacted. This was repeated in waves until the required number of practices was reached. This procedure was intended to exclude selection bias as far as possible. Deviating from the original study protocol, the recruitment period was extended to September 2018 due to recruitment difficulties. In total, 1650 practices were approached. Finally, 114 practices from Mecklenburg-Western Pomerania (n = 54) and Baden-Württemberg (n = 60) were recruited to participate in the nested cRCT. This also shifted the start date of the intervention to the fourth quarter of 2018. In each case, data were retrieved retrospectively with the following measurement points: T0 (4th quarter 2016–1st quarter 2017) and T1 (4th quarter 2017–1st quarter 2018) being baseline, followed by T2 (4th quarter 2018–1st quarter 2019). To realise the planned follow-up, a further point of measurement (T3: 4th quarter 2019–1st quarter 2020) was realised after T2. See Figure 1 for patient and physician/practice flow. 

### 3.3. Randomisation of Practices in cRCT

Practices were randomised into intervention (n = 57) and control (n = 57) by an independent statistician at the Institute for Medical Biometry, University Hospital Heidelberg, Germany. The considered total number of patients (pre-intervention) and region as well as level of urbanization were taken into account and aimed for comparable practices in each arm. 

### 3.4. Interventions

As described above, the regional intervention made use of a public awareness campaign that included web-based activities (https://www.weniger-antibiotika.de/, accessed on 3 May 2023), written information for GPs, posters for waiting rooms, patient magazines, and a plush toy for children. 

The nested cRCT made use of additional elements (Appendix A): Physicians of the intervention group received feedback on practice-specific antibiotic prescribing on two occasions (Appendix A), complemented with an educational outreach visit (Appendix A). During that visit, a specially trained expert discussed practice-specific antibiotic-prescribing feedback. In addition, the practice had access to e-learning modules focusing on strategies for communication with patients expecting a prescription for antibiotics (see TIDieR tables in Appendix A). Patient information on the common cold, middle ear infections, and sinus infections were tailored to different groups of patients, e.g., the elderly or parents of children with ARTIs, and was provided in the form of flyers and information on tablet computers. Information was provided in German, Turkish, Vietnamese, Russian, English, French, and Arabic (Appendix A). Communication designers were involved in the development of the materials.

### 3.5. Data Protection and Consent

All practices, who participated in the cRCT, consented to the use of their claims data and signed a data release form. The practices of the regional intervention were anonymised. As their data are routinely analysed on a legal base, written consent was not required. It is noteworthy that patients were not actively recruited. Cases were automatically included when a respective physician claimed reimbursement from the health insurer. Accordingly, all patients insured with the AOK who consulted a physician in the participating regions for ARTI or cRCT practices, respectively, were included in the study. There was no follow-up in relation to individual patients, so patients may differ from quarter to quarter. 

### 3.6. Outcome Data

Primary outcome measure was the percentage of antibiotic prescribing for ARTIs at T3 (T0 and T1 being baseline), T2 (during intervention for cRCT and regional trial), and T3 (follow-up for cRCT; intervention for regional trial). It was defined as the percentage of patient cases with acute non-complicated infections receiving an antibiotic prescription (ATC codes J01). In this context, we analysed cases with acute bronchitis (age 18–75), sinusitis (>18 years), otitis media (>2 years), upper respiratory tract infection (>1 year), and tonsillitis (>1 year). Diagnoses were based on physician-recorded ICD-10 codes (see Table 6) and prescribing information in the administrative data provided by the health insurer for quarterly reimbursement periods, which were linked by the pseudonymized individual insurance number. Secondary outcomes included the percentage of all observed cases with acute non-complicated infections receiving a (non-recommended) quinolone prescription or (if unavoidable) receiving an antibiotic prescription for guideline-recommended antibiotics (depending on infection) when consulting primary care practices. We used the outcome references of the European Surveillance of Antimicrobial Consumption Network (ESAC-Net) indicators [6], which were tailored to the specifics of CHANGE-3 and coordinated with another study in the same period [14]. 

### 3.7. Sample Size

We expected to find a relative reduction of the percentage of antibiotic prescribing of about 30% (which meant an absolute reduction from 40 to 28%) between the groups. The sample size considerations were based on the chi-squared test using a significance level of α = 5% (two-sided) and a power of (1 − β) = 80%. This resulted in a sample size of n = 244 patients per group. To take the clustered structure into account, we assumed an intra-cluster correlation coefficient (ICC) of 0.2 for patients in practice [20]. This value was based on previous studies. The cluster size was assumed to be m = 80 patients. Thus, the design effect (DE) equaled DE = 1 + (m − 1) × ICC = 1 + 79 × 0.2 = 16.8. These considerations resulted in a sample size of N = n × DE = 244 × 16.8 = 4100 patients per group. Thus, the total sample size was 104 practices of size m = 80, resulting in 8320 patients. A drop-out rate of about 8% was expected, and therefore, we aimed to recruit ten additional practices (see Figure 1 for detailed information on the flow of physicians/practices and patients). The analysis was conducted by using a logistic mixed-effects model, which we expected to explain more of the variance in the prescribing of antibiotics. Therefore, the model was assumed to be more powerful than the chi-squared test. The calculations were done using SAS 9.4 (Proc Power).

### 3.8. Data Collection, Completeness, and Quality

The AOKs are two of the largest statutory health insurance providers, covering about 40% of the statutorily insured in the respective regions. The evaluation was carried out on the basis of two regions that had different prescribing rates in recent years. The endpoints referenced ICD codes recorded in everyday practice as billing data. This means that they are reliable (with the usual limitations). In addition, the inclusion and exclusion criteria were chosen broadly and thus reflect the population. However, all diagnoses are transmitted quarterly without a date.

### 3.9. Statistical Methods

The primary and secondary outcomes, documented data including patient/practices and disease characteristics, and treatment data were first analysed descriptively stratified by intervention arm and timepoint. For categorical variables, absolute and relative frequencies were provided. It should be noted that patient and disease characteristics, treatment data, and practice characteristics differ between outcomes, as the cases considered for each outcome are defined by the respective disease and antibiotic prescription. Logistic mixed-effects models were used to evaluate the primary outcome. The models considered the nested structure of the data with multiple cases per patient and patients nested in practices by including a random effect in the logistic mixed-effects model for patients and practices. As fixed effects, timepoint (by quarter), group (intervention versus control for the cRCT; intervention versus regional intervention, control versus regional intervention in the regional intervention trial), interaction term between time and group, gender, age group (<18, 18–65, >65 years), hierarchical comorbidity groups (0, 1–2, 3–4, ≥5), region (BW, MV), and number of GPs per practice were included as fixed effect. All analyses were performed using R version 4.0.2.

### 3.10. Study Registration and Ethical Approval

The study has been registered with Current Controlled Trials Ltd. with the reference ISRCTN15061174. It was approved by the Ethics Committee of the Rostock University Medical Center in September 2017 (Approval-No. A-2017-0134/ A-2017-0162). The Baden-Württemberg Medical Association (Landesärztekammer) followed this approval with its ethics vote (B-F-2017-114). 

## 4. Discussion

### 4.1. Summary of Findings

In view of a general trend towards the restrained use of antibiotics in German primary care, the regression analysis confirmed a reduction in the percentage of antibiotic prescribing over time from T0 to T3. This trend was evident in both the control and the intervention group; a significant difference between both groups, however, was not present. Thus, no effect of the implementation program was found in the present study. At the same time, in routine care (which was exposed to the public awareness campaign), the percentage of antibiotic prescribing was higher than in both groups of the cRCT. This suggests that the GPs who participated in the study are more sensitive to the topic. With regard to secondary outcomes, the data analysis shows that in the nested cRCT, the prescribing of quinolones was reduced, and the proportion of guideline-recommended antibiotics increased. This is an important result with regard to the development of antibiotic resistance.

### 4.2. Findings in Research Context

Previous intervention studies on the reduction of antibiotics in ARTIs found that intervention elements such as exchange with colleagues, the doctor–patient conversation, and prescription feedback can significantly reduce the prescribing of antibiotics in countries with medium antibiotic prescribing rates, such as Germany [14,15,16,17]. With regard to the influence of the practice team on the percentage of antibiotic prescribing, our results are similar to previous findings [13,14,18]. Our study did not find any positive effects of including the practice team in the intervention in terms of antibiotic prescribing. Moreover, no significant effect of the public campaign on the percentage of prescribing was demonstrated. However, this result contradicts other regional and national experiences [7]. Possible reasons for this result could be the focus on digital content of the public awareness campaign, as well as a possibly insufficient reach in the targeted population. The process evaluation in CHANGE-3 clearly showed that digital content was used less in German primary care compared to print material [21]. Hence, future campaigns should carefully assess appropriate formats and consider related potential during- and post-pandemic developments. The learning effect in quality circles might be greater (as in the ARena study [14]) than that achieved by outreach visits [13]. 

### 4.3. Strengths and Limitations

The study design allowed us to investigate to what extent intervention elements of the cRCT were equivalent, superior, or inferior to a public awareness campaign. In addition, digital formats (webpage and digital patient information content provided via tablet PCs) were used here on a large scale, and their effect on the reduction of antibiotic prescribing was tested. The study thus makes a contribution to research into effective intervention approaches in the German primary care setting. A comprehensive process evaluation also provided important insights into the implementation and use of the intervention elements. This is highly relevant for future studies.

The results of this study are limited in that it can be assumed that the majority of participants are particularly highly motivated to optimise their antibiotic prescribing any-way. Moreover, with the inclusion of patients of the AOK, only a part of the actual patients participated in the study. However, with 40% of all insured persons, the AOK is the largest German health insurer. It needs to be mentioned that the effect of reducing antibiotic prescriptions by 30% originally assumed for the study was not achieved. This must also be considered against the background of a trend toward the reduction of antibiotic prescriptions in Germany that was not foreseeable in previous years. Achieving a pronounced reduction during this positive development is difficult. Regardless of this, the number of cases in the study is sufficient to provide statistically adequate evidence of the effect observed here.

### 4.4. Implications for Practice

Even though the tested intervention did not lead to a significant reduction in antibiotic prescriptions for ARTIs, it has been shown that we still have scope for a further reduction in antibiotic prescribing in German primary care. The same applies to the adequate use of broad-spectrum antibiotics. For primary care, this means that the advantages and disadvantages of each antibiotic prescription should be weighed.

### 4.5. Future Research 

The results of the CHANGE-3 study indicate group-specific antibiotic prescribing behaviour. Women, patients over 65 years of age, and patients with a higher number of chronic diseases were more likely to receive an antibiotic than the respective other groups. Qualitative studies should investigate why these groups receive antibiotics more frequently in comparison. Future interventional approaches should also focus more on these patient groups and develop and test target group-specific approaches. It should be taken into account that digital content receives less attention than printed materials. To what extent the digitalisation of the Corona pandemic has an influence here will only become clear in the coming years. It is likely that developments have taken place in the meantime that favour the use of digital content.

## 5. Conclusions

The CHANGE-3 study aimed to investigate how antibiotic prescribing for non-complicated ARTIs in primary care can be reduced to a reasonable level in two regional areas of Germany. Although a reduction in antibiotic prescribing can be demonstrated in both groups of the cRCT, the two groups do not differ significantly from each other. The comparison with the group of practices that were not included in the cRCT but may have been influenced by the public campaign also showed no significant differences. It can be assumed that the tested intervention elements had no effect on prescribing of antibiotics in ARTIs. However, the data analysis shows that in the nested cRCT, the prescribing of quinolones could be reduced, and the proportion of guideline-recommended antibiotics increased. This is a promising result with regard to the development of antibiotic resistance.

## Figures and Tables

**Figure 1 antibiotics-12-00850-f001:**
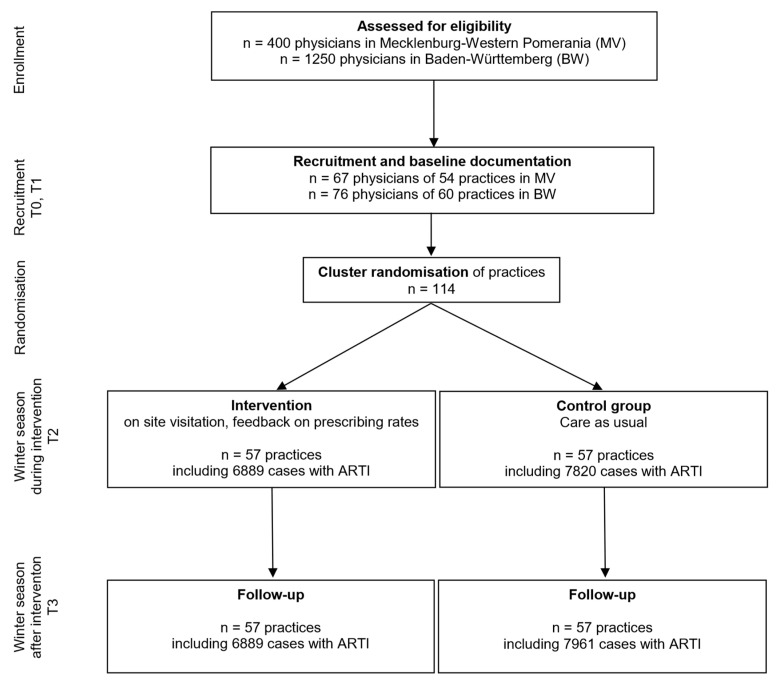
Patient and physician flow in the nested cluster randomised controlled trial.

**Table 1 antibiotics-12-00850-t001:** Baseline data of cRCT, including 61,390 cases from 114 practices.

Group	Intervention				Control			
Time	T0	T1	T2	T3	T0	T1	T2	T3
Cases	n = 6481 (%)	n = 7269	n = 6889	n = 6769	n = 8903	n = 9298	n = 7820	n = 7961
Cases from BW versus MV	4601 (71.0)	4999 (68.8)	4883 (70.9)	4702 (69.5)	6483 (72.8)	6476 (69.6)	5397 (69.0)	5550 (69.7)
Age (patients)								
- <18	622 (9.6)	696 (9.6)	652 (9.5)	631 (9.3)	901 (10.1)	974 (10.5)	893 (11.4)	857 (10.8)
- 18–65	5121 (79.0)	5827 (80.2)	5593 (81.2)	5562 (82.2)	7158 (80.4)	7479 (80.4)	6315 (80.8)	6529 (82.0)
- >65	738 (11.4)	746 (10.3)	644 (9.3)	576 (8.5)	844 (9.5)	845 (9.1)	612 (7.8)	575 (7.2)
Female (versus male patients)	3449 (53.2)	3860 (53.1)	3617 (52.5)	3609 (53.3)	4734 (53.2)	4860 (52.3)	4070 (52.0)	4134 (51.9)
Number of hierarchised morbidity groups								
- 0	3245 (50.1)	3658 (50.3)	3590 (52.1)	3548 (52.4)	4625 (51.9)	4968 (53.4)	4373 (55.9)	4561 (57.3)
- 1, 2	1901 (29.3)	2196 (30.2)	1964 (28.5)	1999 (29.5)	2664 (29.9)	2696 (29.0)	2209 (28.2)	2239 (28.1)
- 3, 4	789 (12.2)	836 (11.5)	755 (11.0)	703 (10.4)	931 (10.5)	918 (9.9)	707 (9.0)	682 (8.6)
- ≥5	546 (8.4)	579 (8.0)	580 (8.4)	519 (7.7)	683 (7.7)	716 (7.7)	531 (6.8)	479 (6.0)
Disease								
- ARTI	4837 (74.6)	5637 (77.5)	5363 (77.8)	5438 (80.3)	6881 (77.3)	7630 (82.1)	6321 (80.8)	6580 (82.7)
- Bronchitis	1435 (22.1)	1381 (19.0)	1229 (17.8)	1133 (16.7)	2002 (22.5)	1902 (20.5)	1379 (17.6)	1379 (17.3)
- Tonsillitis	357 (5.5)	359 (4.9)	393 (5.7)	335 (4.9)	381 (4.3)	305 (3.3)	326 (4.2)	308 (3.9)
- Sinusitis	203 (3.1)	202 (2.8)	200 (2.9)	174 (2.6)	662 (7.4)	594 (6.4)	602 (7.7)	612 (7.7)
- Otitis	344 (5.3)	355 (4.9)	338 (4.9)	290 (4.3)	328 (3.7)	297 (3.2)	281 (3.6)	262 (3.3)
Antibiotic prescription	1537 (23.7)	1462 (20.1)	1113 (16.2)	1020 (15.1)	1585 (17.8)	1356 (14.6)	1067 (13.6)	991 (12.4)
One GP per practice (versus two or more)	4319 (66.6)	4388 (60.4)	3941 (57.2)	3735 (55.2)	4748 (53.3)	4887 (52.6)	4148 (53.0)	4149 (52.1)

Note: BW = Baden-Württemberg. MV = Mecklenburg-Western Pomerania. ARTI = acute respiratory tract infections. Since GPs assign more than one ICD-10 code in some cases, more ICD-10 diagnoses are shown in the table than cases.

**Table 2 antibiotics-12-00850-t002:** Logistic mixed regression model of cRCT data for the primary outcome (percentage of cases with index diagnosis and antibiotic prescribing; over time, T0 to T3).

Antibiotic Prescription	OR	95% CI	*p*-Value
Over Time (intervention and control)	0.836	0.809; 0.863	<0.001
Intervention versus control	1.285	0.874; 1.889	0.202
Female versus male	1.279	1.213; 1.349	<0.001
Age < 18 versus 18–65 years	0.808	0.733; 0.89	<0.001
Age > 65 versus 18–65 years	1.813	1.666; 1.972	<0.001
Over time: Intervention versus control *	0.958	0.915; 1.002	0.063

Note: * Interaction between time and group.

**Table 3 antibiotics-12-00850-t003:** Logistic mixed regression model of the primary outcome (percentage of cases with index diagnosis and antibiotic prescription; over time (T0 to T3); adjusted for sex, age, number of hierarchised morbidity groups, state, practice type).

Antibiotic Prescription	OR	95% CI	*p*-Value
Over time (intervention and control)	0.842	0.815; 0.87	<0.001
Intervention versus control	1.253	0.859; 1.827	0.242
Female versus male	1.26	1.195; 1.329	<0.001
Age < 18 versus 18–65 years	0.938	0.849; 1.035	0.204
Age > 65 versus 18–65 years	1.297	1.176; 1.429	<0.001
Number of hierarchised morbidity groups 1,2 versus 0	1.334	1.254; 1.419	<0.001
Number of hierarchised morbidity groups 3,4 versus 0	1.568	1.434; 1.715	<0.001
Number of hierarchised morbidity groups 5+ versus 0	2.046	1.835; 2.281	<0.001
Region Baden-Württemberg versus Mecklenburg-Western Pomerania	1.044	0.72; 1.513	0.822
Single versus group practice	0.817	0.718; 0.929	0.002
Over time: Intervention versus control *	0.956	0.913; 1.001	0.053

Note: * Interaction between time and group.

**Table 4 antibiotics-12-00850-t004:** Baseline data of the “care as usual” group (public campaign), 4,070,998 cases.

	T0	T1	T2	T3
Number of cases	(n = 1,042,485), (%)	(n = 1,069,389)	(n = 956,707)	(n = 1,002,417)
Patients from BW versus MV	972,477 (93.3)	994,223 (93.0)	896,799 (93.7)	946,273 (94.4)
Age groups of patients				
- <18	97,516 (9.4)	93,461 (8.7)	83,313 (8.7)	82,208 (8.2)
- 18–65	855,942 (82.1)	892,083 (83.4)	805,367 (84.2)	855,017 (85.3)
- >65	89,027 (8.5)	83,845 (7.8)	68,027 (7.1)	65,192 (6.5)
Female patients (versus male)	546,166 (52.4)	550,301 (51.5)	488,030 (51.0)	512,457 (51.1)
Number of hierarchised morbidity groups				
- 0	566,854 (54.4)	590,243 (55.2)	537,484 (56.2)	564,413 (56.3)
- 1, 2	311,461 (29.9)	314,149 (29.4)	276,237 (28.9)	291,900 (29.1)
- 3, 4	103,293 (9.9)	103,320 (9.7)	89,231 (9.3)	91,951 (9.2)
- ≥5	60,877 (5.8)	61,677 (5.8)	53,755 (5.6)	54,153 (5.4)
One GP per practice (versus two or more)	564,936 (54.2)	580,523 (54.3)	519,485 (54.3)	540,306 (53.9)
Disease (patients with diagnosis)				
- ARTI	766,094 (73.5)	809,997 (75.7)	729,924 (76.3)	783,362 (78.1)
- Bronchitis	268,274 (25.7)	258,468 (24.2)	209,589 (21.9)	206,242 (20.6)
- Tonsillitis	66,873 (6.4)	63,712 (6.0)	60,519 (6.3)	59,158 (5.9)
- Sinusitis	54,209 (5.2)	47,136 (4.4)	46,410 (4.9)	40,756 (4.1)
- Otitis	34,251 (3.3)	33,619 (3.1)	31,542 (3.3)	30,432 (3.0)
Antibiotic prescription	267,885 (25.7)	259,890 (24.3)	213,161 (22.3)	201,495 (20.1)

Note: BW = Baden-Württemberg. MV = Mecklenburg-Western Pomerania. ARTI = acute respiratory tract infections. Since GPs assign more than one ICD-10 code in some cases, more ICD-10 diagnoses are shown in the table than cases.

**Table 5 antibiotics-12-00850-t005:** Logistic mixed regression model of the primary outcome—percentage of patients with index diagnosis and antibiotic prescription—over time (T0 to T3), cRCT (intervention group and control group) to the regional intervention.

Antibiotic Prescription	OR	95% CI	*p*-Value
Time	0.903	0.901; 0.905	<0.001
cRCT control versus care as usual	0.634	0.489; 0.821	0.001
cRCT intervention versus care as usual	0.768	0.589; 1.002	0.052
Female versus male	1.241	1.235; 1.247	<0.001
Age < 18 versus 18–65	0.929	0.92; 0.937	<0.001
Age > 65 versus 18–65	1.24	1.228; 1.252	<0.001
Number of hierarchised morbidity groups 1,2 versus 0	1.251	1.244; 1.258	<0.001
Number of hierarchised morbidity groups 3,4 versus 0	1.57	1.556; 1.583	<0.001
Number of hierarchised morbidity groups 5+ vsersus 0	1.739	1.719; 1.759	<0.001
BW versus MV	0.797	0.746; 0.852	<0.001
Single versus group practice	0.831	0.82; 0.843	<0.001
Time: Intervention group versus care as usual *	0.946	0.918; 0.974	<0.001
Time: Control group versus care as usual *	0.913	0.886; 0.94	<0.001

Note: BW = Baden-Württemberg. MV = Mecklenburg-Western Pomerania. * Interaction between time and group.

**Table 6 antibiotics-12-00850-t006:** ICD-10 codes used to identify patients with index disease.

Infections	ICD-10
Acute bronchitis	J20
Acute bronchitis	J21.0
Acute bronchitis	J21.1
Acute bronchitis	J21.8
Acute bronchitis	J21.9
Acute bronchitis	J22
Acute bronchitis	J40
Acute upper respiratory tract infection	J00
Acute upper respiratory tract infection	J02.0
Acute upper respiratory tract infection	J02.8
Acute upper respiratory tract infection	J02.9
Acute upper respiratory tract infection	J04
Acute upper respiratory tract infection	J06
Acute upper respiratory tract infection	J10.1
Acute upper respiratory tract infection	J11.1
Sinusitis	J01
Tonsillitis	J03.0
Otitis media	H65.0
Otitis media	H65.1
Otitis media	H65.9
Otitis media	H66.0
Otitis media	H66.4
Otitis media	H66.9

## Data Availability

Data and materials are available upon request.

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
