# Peer review of "Optimizing Antibiotic Prescribing for Acute Respiratory Tract Infections in German Primary Care: Results of the Regional Intervention Study CHANGE-3 and the Nested cRCT"

_antibiotics, 2023, doi:10.3390/antibiotics12050850_

Round 1
Reviewer 1 Report
Dear authors,
I will first give some general comments, then some major and minor comments that more or less follow the chronology of the paper.
General comments
I find the introduction too long and suggest focusing to what is known from other intervention studies, and how the GHANGE-3 study could contribute to our knowledge on this important and complex area of clinical practice.
You state that the aim of the study is to investigate how antibiotic prescribing for non-complicated ARTI can be reduced to a reasonable level. What is a reasonable level in this context? Is the definition of non-complicated ARTI solely based on ICD-10 codes? Was the use of codes uniform over practices? Have you considered diagnostic drift during the cRCT?
You present data on the proportion of patients receiving antibiotics. Do you have data on dose and treatment duration?
How is the use of rapid diagnostic tests in these practices? Any change over the period?
Use of terminology. Rate vs. proportions. In line 242 you state that the primary outcome measure is the prescribing rate. However, in the results you present proportions. In line 251 you state that secondary outcomes included the percentage of all observed cases – which in my opinion is the correct terminology. This is in fact the periodic prevalence of antibiotic prescribing. This should be addressed simultaneously with repeated prescribing to individuals. How was the distribution of multiple prescriptions in the two groups? Any change over the period?
Is the areas in BW and MW comparable when it comes to social deprivation?
You mix the spelling of the region Baden-Wuerttemberg with Baden-Württemberg. I suggest you use the German names.
Major comments
You approached 1,650 practices of which 114 agreed to participate in the cRCT. It is unclear to me whether these practices may represent some kind of selection. You scarcely comment on this in the discussion (l. 313-14). Can you elaborate on this topic?
Tables 1&4 are in my opinion quite messy and some parts are unnecessary detailed. Other parts could benefit on more detail. I have the following suggestions:
- Use BW for Bad-Württemberg
- Avoid % in the brackets. Explain this in the first column
- Consider using 10-year age-groups. The 18-65y is quite big compared to the others
- Consider reducing number of co-morbidity groups (0, 1-2, ≥3). It is well known that antibiotic prescribing increases with increasing number of comorbidities. This will also affect the outputs in tb. 3&5.
- Consider cutting the information on DMP. You do not use this in your regression model, and the differences groups and time-points are minor
- Tab. 1. For Intervention group T0 there are 6481 cases. In the part of the table that denotes “Disease (patients)” the total number is 7176. Neither do the number add up in the other columns. This needs an explanation. Also – the numbers for sinusitis are not in bold. Any reason for this (same in tab. 4)?
- For tab. 4. Consider revision according to my suggestions for tab. 1
Your sample size calculation assumes a 30% difference between groups. The observed difference is far less. Possible consequences should be addressed in the discussion.
It is unclear to me how the cases in the public campaign are identified. How many practices contributed data? Any diagnostic drift over the period?
Minor comments
This is a highly complex intervention, and you present a fairly narrow set of outcomes. Have you considered whether there may be patient groups that benefitted more from the cRCT?
I think that the result section would benefit from explicit statements on the unit of analysis.
In your section 4.3 you state that your study makes an important contribution to research into effective intervention approaches in German primary care. I care to disagree. You found no effect of the cRCT (possibly due to a very broad aggregation of data and heterogeneity between groups), and it is unclear to me how this study contributes to new insight on the topic. However, I agree on the need to reduce the consumption of antibiotics in Germany, and I think a mix of public campaign and outreach to all practices are important parts of the armamentarium.
Author Response
Dear Reviewer 1,
thank you very much for your constructive feedback, which we were very happy to take into account when revising the manuscript.
- according to your suggestion, we have shortened the introduction or summarised aspects of it.
- In section 3.6 we have included Table 6, which provides an overview of all ICD-10 codes used to identify cases of ARTI in the routine dataset. As described in section 3.6, ARTIs were identified based on these ICD-10 codes and with cuts in the age of patients (e.g. exclusion of children under 2 years of age in otitis media; due to specified guideline recommendation). The use of ICD-10 codes is always associated with some variability (but this is the same problem for intervention and control). However, no alternative exists when using claims data from German health care.
When planning the study in 2016, we assumed a 30% reduction in antibiotic prescriptions in the intervention group. Previous studies have achieved up to 40% in the German setting. We did not reach this target in the study, which is also due to the fact that the prescription figures in Germany have steadily decreased in recent years. A reduction in the order of magnitude is thus becoming increasingly difficult; however, we did not foresee this development in 2016.
- The German routine data do not provide information on the applied dosage or duration of antibiotic use. We therefore focused purely on the decision for prescription of an antibiotic in the study.
- Since diagnostic tests are not billable via the statutory health insurance in Germany (and are therefore private services), corresponding information on their use is not recorded in the routine data. We are therefore unable to make any statements on this.
- You are right; thank you for pointing that out. It is not a rate but the percentage of antibiotic prescriptions (ATC J01) in the cases considered (ICD-10 codes, see Table 6). We have revised this accordingly in the manuscript.
With regard to multiple prescription of antibiotics, the routine data are unfortunately inaccurate because they provide data on a quarterly basis. Exact tracking of individual patients was not possibly for technical reasons and was not the aim of our study.
- MV and BW are not comparable with each other. While BW is in western Germany, MV is in eastern Germany. Already historically, there are numerous differences (but also with regard to social differences). Our aim was to include two regions that exhibit precisely these differences. In eastern Germany, for example, fewer antibiotics were prescribed historically. We also wanted to know to what extent our intervention concept works in both parts of Germany and leads to reductions in the prescription of antibiotics.
- As suggested, we use the German term Baden-Württemberg in the revised version. Thank you for the advice.
- For recruitment, all GPs resident within a 100 km radius of Heidelberg (BW) and Rostock (MV) were identified from the complete register of all GPs in each of the two federal states. Random samples were drawn from this population and the selected GPs were contacted. This was repeated in waves until the required number of practices was reached. This procedure was intended to exclude selection bias as far as possible.
We have clarified this in the section on recruitment (3.2.).
- In tables 1 and 4, we have abbreviated BW according to the recommendations, removed the percentage signs from the columns, and in table 1 we have also removed the information on the DMP. The information on the diagnoses is correct, because some doctors code more than one ICD-10 diagnosis for a patient. We have now placed this information as a note under the table. We have not implemented the recommendation to structure the age groups more finely and to summarise the information on the number of comorbidities. Both groupings are based on internationally used classifications, which can also be found in the literature.
- We have addressed the comment on the case number calculation, which assumes an antibiotic reduction of 30%, in the discussion under "Strengths and limitations". Despite the fact that we were not able to prove this reduction in the study, we included enough cases to be able to make statistically valid statements for the observed effect.
- The data basis was identical for all study groups. We have added to this in section 3.1 Study design.
- The analysis (e.g. Table 2 and description of the results in section 2.2. for the cRCT and Table 5 for public campaign) shows that e.g. women and patients over 65 have a higher risk of being prescribed an antibiotic, as do multimorbid patients. These are of course also the groups of people who would particularly benefit from a reduction in their prescription risk. Our subgroup analyses focused on the prescription of recommended antibiotics for the index diseases under consideration. The corresponding results are presented in section 2.5 and show how the prescription of quinolones in cRCT decreases while the share of recommended antibiotics increases.
- In the results section, we now write uniformly of the percentage of antibiotic prescriptions (instead of rate).
- Thank you for your feedback. We have changed "important contribution" to "contribution" in section 4.3.
We uploaded a version of the manuscript with change mode and a clean version.
Reviewer 2 Report
Dear authors
This is a crucially important manuscript which has described problems of the issue of blowing antibiotic resistance which has affected infections eradication. The manuscript has been written well and some minor improvements are required.
The authors can propose control strategies and combination therapies with this regard. I also propose to include the efficacy of drug delivery systems in the discussion section such as the following manuscript:
Nickel Nanoparticles: Applications and Antimicrobial Role against Methicillin-Resistant Staphylococcus aureus Infections
kind regards
Dear editor
This is a crucially important manuscript which has described problems of the issue of blowing antibiotic resistance which has affected infections eradication. An important topic has been studied and results of this clinical trial are appropriate. The manuscript has been written well and some minor improvements are required.
The authors can propose control strategies and combination therapies with this regard. I also propose to include the efficacy of drug delivery systems in the discussion section such as the following manuscript:
Nickel Nanoparticles: Applications and Antimicrobial Role against Methicillin-Resistant Staphylococcus aureus Infections
kind regards
Author Response
Reviewer 2
Thank you for your reference to the publication. The research field around the development of nickel nanoparticles is important, however we can hardly establish a connection to our study. In Germany, MRSA mainly play a role in the inpatient sector, much less in outpatient care, which is our focus. We are also looking at how care can be optimised. The use of drugs or medical devices does not play a role here. We have therefore decided not to cite the study mentioned and hope for your understanding.
We uploaded a version of the manuscript with change mode and a clean version.
Reviewer 3 Report
Line 43 in introduction section 'nevertheless in Germany, 40% of all ARTI cases seen in primary care are still treated with antibiotics' needs appropriate referencing.
Quality of tables (1-4) should be improved.
English Language is fine throughout the manuscript
Author Response
REVIEWER 3
Thank you very much for your comments. References 6 and 7 refer to the prescription of 40% antibiotics in ARTI in Germany. The following figures on Southern Europe etc. have the same references, which is why we had only quoted them after the following sentence. To make this clearer, however, we have now added the references after the sentence on the German figures.
Tables 1 and 4 have been revised. From your comments we could not conclude which revision you propose for tables 2 and 3. We have tried to optimise them as best we can.
We uploaded a version of the manuscript with change mode and a clean version.
Reviewer 4 Report
This is a study examining the effect of intervention to reduce the use of community antibiotics. this is an important and admirable goal.
there are several issues with this manuscript that need to be addressed before this is ready for publication:
1. Tables description is too short and not clear, the foot note is missing describing what the abbreviations mean.
2. the actual data of prescribing different antibiotics is missing, what was the rate of prescription? for each antibiotic and the overall rate per period. this should be shown either as a table or as a graph
3. antibiotic data needs to be presented as prescription per 1,000 doctor visits to control for the different sizes of practice, this is the accepatable manner to present data.
4. the stats used are unacceptable for this type of study, what should have been used is an interrupted time series analysis to examine the correlation between the intervention and the use of antibiotics (ARIMA). further details can be found on cochrane or shea website for example
https://learningce.shea-online.org/system/files/Assessing%20Impact%20of%20Stewardship%20The%20Why%2C%20When%2C%20and%20How%20of%20Interrupted%20Time%20Series%20-%20Jessina%20McGregor%2C%20PhD.pdf
5. the introduction is too long
6. the English needs work
7. the authors describe the effect on quinolones, was there a "squeeze the balloon effect?
8. the intervention is not described clearly, what is the point of publishing this intervention if others cannot utilized it.
9. the manuscript does not follow the acceptable publication guidelines as stated by the "strobe-ams" document. https://bmjopen.bmj.com/content/6/2/e010134
the english needs some work
Author Response
Dear Reviewer 4,
thank you very much for your constructive feedback, which we were very happy to take into account when revising the manuscript. We made the following changes:
- Tables 1 and 4 in particular have been revised; headings have been adjusted for the other tables.
- Routine data were evaluated for the data analysis. Only quinolones were analysed separately, otherwise all antibiotics under the ATC code J01 were combined for the analysis (see methods section). Therefore, a listing of the individual antibiotics is not possible. Although it is also possible to examine the prescription of individual antibiotics with the German routine data, this was not the question addressed in our study. Accordingly, we did not receive this data.
- In the German routine data, we generally do not have any information on patient visits. A comparison of prescriptions - visits is not possible. We therefore compare the percentage of antibiotic prescriptions with patient cases. We control the size of the practices in the analysis with the help of the proxy "1 GP versus 2 or more GPs".
- Many thanks for this valuable comment. The analysed claims data have several limitations that make the analysis using interrupted time series challenging. The time points are not distributed equidistant, only Q4 and Q1 data is available, the complex intervention was introduced sequentially and not in all regions/practices at the same time point, and the time after intervention was short caused by a limited project time.
- The introduction has been shortened.
- We have hired a translation agency to re-edit the manuscript for English.
- Developments throughout Germany argue against a "squeeze the balloon effect". In recent years, the prescription rate of antibiotics has fallen in Germany as a whole. We also see a decline in the public campaign group and in the control group (Table 1 and 4).
- All six components of the intervention are described in detail in Supplement 1 using the TIDieR tables. There you will find a total of 6 pages of information on each component in terms of objectives, material used, implementation, providers, place of implementation, timing, etc.
- It is true that we did not follow the STROBE Statement. Since we conducted a cluster-randomised controlled trial, we followed the CONSORT Statement for cRCTs.
We uploaded a version of the manuscript with change mode and a clean version.
Round 2
